# Comparison of Climate Change Effects on Wheat Production under Different Representative Concentration Pathway Scenarios in North Kazakhstan

Zhanassyl Teleubay [1,*], Farabi Yermekov [2], Arman Rustembayev [2], Sultan Topayev [2], Askar Zhabayev [2], Ismail Tokbergenov [2], Valentina Garkushina [3], Amangeldy Igilmanov [3], Vakhtang Shelia [4] and Gerrit Hoogenboom [4]

[1] Department of Geography and Atmospheric Sciences, The Ohio State University, Columbus, OH 43210, USA
[2] Center for Technological Competence in the Field of Digitalization of the Agro-Industrial Complex, S.Seifullin Kazakh Agrotechnical Research University, Astana 010000, Kazakhstan
[3] Faculty of Land Management, Architecture and Design, S.Seifullin Kazakh Agrotechnical Research University, Astana 010000, Kazakhstan
[4] Department of Agricultural and Biological Engineering, Global Food Systems Institute, University of Florida, Gainesville, FL 32611, USA
[*] Correspondence: teleubay.1@buckeyemail.osu.edu or zhanassyl.kz@gmail.com

**Abstract:** Adverse weather conditions, once rare anomalies, are now becoming increasingly commonplace, causing heavy losses to crops and livestock. One of the most immediate and far-reaching concerns is the potential impact on agricultural productivity and global food security. Although studies combining crop models and future climate data have been previously carried out, such research work in Central Asia is limited in the international literature. The current research aims to harness the predictive capabilities of the CRAFT (CCAFS Regional Agricultural Forecasting Toolbox) to predict and comprehend the ramifications stemming from three distinct RCPs, 2.6, 4.5, and 8.5, on wheat yield. As a result, the arid steppe zone was found to be the most sensitive to an increase in greenhouse gases in the atmosphere, since the yield difference between RCPs 2.6 and 8.5 accounted for almost 110 kg/ha (16.4%) and for 77.1 kg/ha (10.4%) between RCPs 4.5 and 8.5, followed by the small hilly zone with an average loss of 90.1 and 58.5 kg/ha for RCPs 2.6–8.5 and RCPs 4.5–8.5, respectively. The research findings indicated the loss of more than 10% of wheat in the arid steppe zone, 7.6% in the small hilly zone, 7.5% in the forest steppe zone, and 6% in the colo steppe zone due to climate change if the modeled RCP 8.5 scenario occurs without any technological modernization and genetic modification. The average wheat yield failure in the North Kazakhstan region accounted for 25.2, 59.5, and 84.7 kg/ha for RCPs 2.6–4.5, 4.5–8.5, and 2.6–8.5, respectively, which could lead to food disasters at a regional scale. Overall, the CRAFT using the DSSAT crop modeling system, combined with the climate predictions, showed great potential in assessing climate change effects on wheat yield under different climate scenarios in the North Kazakhstan region. We believe that the results obtained will be helpful during the development and zoning of modified, drought-resistant wheat varieties and the cultivation of new crops in the region.

**Keywords:** agriculture; food security; crop modeling; RCP; wheat yield prediction

## 1. Introduction

The global community has awakened to a looming crisis transcending borders, economies, and ecosystems in recent decades: climate change [1,2]. As atmospheric greenhouse gas concentrations continue to rise, the planet's climate is undergoing rapid transformations, triggering a cascade of effects that ripple across diverse sectors of society [3]. One of the most immediate and far-reaching concerns is the potential impact on agricultural productivity and global food security [4,5]. Studies of such influence have been

conducted for many countries, reaching conclusions on the complexity of the situation and the requirement of particular policies to mitigate the consequences [6–9]. As a cornerstone of human civilization, agriculture sustains livelihoods, economies, and societies worldwide. Yet, this intricate web of sustenance is inextricably tied to climatic conditions. Rising temperatures, altered precipitation patterns, and changing atmospheric carbon dioxide levels challenge agricultural systems' stability and predictability [10–13]. The consequences of such shifts can potentially disrupt the delicate balance between food production and population growth, compromising our ability to feed a burgeoning global population [14].

Extreme weather events, once rare anomalies, are now becoming distressingly familiar, exacting heavy tolls on crops and livestock [15–17]. Prolonged droughts, unpredictable rainfall, and intensified storms disrupt planting cycles and reduce yields [18–20]. The nexus between climate change and food security is multifaceted. It extends beyond the fields to impact post-harvest storage, transportation, and distribution systems [21]. These disruptions disproportionately affect vulnerable communities, particularly in developing nations, given their reliance on subsistence agriculture and limited adaptive capacity [22]. The global food trade, a lifeline for many countries, faces disruptions as production centers shift and supply chains strain under pressure [23,24]. These mounting concerns necessitate urgent action, not only in mitigating the drivers of climate change but also in enhancing the resilience of agricultural systems to its impacts.

Numerous models and software tools have been developed to predict crop yields in response to temperature changes and other environmental factors. These tools use a combination of climate data, agronomic knowledge, and mathematical models to estimate the potential crop production. Here are a few notable ones:

1. DSSAT (Decision Support System for Agrotechnology Transfer) 4.7.5 is a widely used program that simulates the growth of crops and their response to various management practices, including temperature changes. It includes a suite of crop models that can affect the impact of temperature, rainfall, and other factors on crop yields. Xiang et al. [25] used the DSSAT model in combination with the MODFLOW groundwater flow model to facilitate assessments of irrigation technology changes, crop choices, and strategies for adapting to climate change in various regions. Mubeen et al. [26] adapted the DSSAT model to determine the impact of climate change through the elevated $CO_2$ condition. The authors suggest that cultivating wheat and cotton varieties with high water use efficiency could be pivotal in sustaining crop production. Attia et al. [27], following the DSSAT calibration and evaluation algorithms for maize cultivation, believe that compost application with retained crop residues is a promising strategy for enhancing agronomic outcomes and environmental sustainability in maize cultivation on arid soils.

2. APSIM (Agricultural Production Systems sIMulator) is another comprehensive software tool that models various aspects of agricultural systems, including crop growth, soil processes, and climate interactions. It can be used to assess the impact of temperature changes on crop yields and inform adaptation strategies. Vogeler et al. [28] found that for well-drained soils in regions with high precipitation and no water limitations, the APSIM model displays low sensitivity to soil hydraulic parameters and suggests that general data from databases may be justifiable instead of relying solely on site-specific measurements of hydraulic properties. Research by Wimalasiri et al. [29] highlights the potential of selecting specific cultivars and adjusting planting dates as climate change adaptation strategies based on APSIM crop model simulation results.

3. STICS (Simulateur mulTIdisciplinaire pour les Cultures Standard) is a crop model for simulating various crops and cropping systems. It considers temperature, precipitation, and other environmental factors to predict crop growth and yield [30]. Fraga et al. [31] adapted the STICS model to suit the unique conditions of Portuguese wine growing and its diverse grapevine varieties. The authors assume that the STICS

model holds potential as a decision-support tool for both short- and long-term strategic planning in the Portuguese viticulture sector.

4.  Global Yield Gap Atlas (GYGA): While not a standalone piece of software, the Global Yield Gap Atlas provides an online platform where users can access global and regional data on actual and potential crop yields. It offers insights into the yield gaps that exist and how they might change under different scenarios, including temperature changes. In the research of Grassini et al. [32], the authors followed the idea of achieving maximum yield potential under sustainable usage of water resources and natural ecosystem protection. It was found that GYGA successfully estimated yield potential, yield gaps, and water productivity for 13 crops across 70 countries around the world [33].

As we transition from evaluating various crop models for agricultural simulations, it is imperative to consider the broader environmental context. Climate change is a pivotal factor that can significantly impact crop growth and yield [34]. Current research includes wheat yield estimation under three different Representative Concentration Pathway (RCP) scenarios defined by the Intergovernmental Panel on Climate Change (IPCC) for the year 2050 [35,36]. RCP scenarios offer a spectrum of future climate projections, ranging from stringent mitigation efforts in RCP 2.6 to business-as-usual emissions in RCP 8.5.

Current research aims to harness the predictive capabilities of the CRAFT v3.4 (CCAFS Regional Agricultural Forecasting Toolbox) [37] to forecast and comprehend the ramifications stemming from three distinct RCPs on wheat yield in the North Kazakhstan region [38]. The CRAFT v3.4 is a multifunctional software platform that can run pre-installed crop models from DSSAT, SARRA-H, and APSIM. By employing the CRAFT, we aim to unravel the intricate relationship between varying climatic conditions, as delineated by RCP 2.6, 4.5, and 8.5 scenarios, and the resulting alterations in wheat yield. This endeavor seeks to foresee potential changes in yield and deepen our understanding of the underlying mechanisms driving these shifts, thereby contributing to a more robust comprehension of the dynamic interplay between climate change and agricultural productivity [39]. Wheat is a staple crop and a significant contributor to Kazakhstan's economy and the region's food security [40]. Given the country's vast geographical span and diverse climatic conditions, understanding how climate scenarios, such as those represented by RCPs 2.6, 4.5, and 8.5, could affect wheat production is crucial. The combination of rising temperatures, altered precipitation patterns, and increased water stress could profoundly impact wheat production in Kazakhstan [41,42].

Research based on the three RCP scenarios for modeling wheat yield in the North Kazakhstan Region for the year 2050 provides an opportunity to assess the impact of climate change on agriculture in the Republic of Kazakhstan and, accordingly, on the country's and Central Asia's food security. According to the USDA's Global Agricultural Information Network report, Kazakhstan harvested 16,404 million tons of wheat in 2022 and exported almost half of it (47%) [43]. Since the top five importers were Uzbekistan, Afghanistan, Tajikistan, Iran, and Turkmenistan, we can firmly state Kazakhstan's high importance in Central Asia's food security. If preventive and adaptive measures are not taken, about 80 million people in the region could be at risk of food insecurity.

This paper is organized as follows: (1) the materials and methods used in the current research for assessing the effects of climate change on wheat production in the North Kazakhstan region are explained, (2) the results of the simulations are presented, (3) the advantages, disadvantages, limitations, and possibilities of the work are discussed, and (4) conclusions and recommendations for future research are provided.

## 2. Materials and Methods

### 2.1. Study Area

The North Kazakhstan Region occupies the southern outskirts of the West Siberian Plain and a part of the Kazakh Uplands (Sary-Arka). The territory of the region covers

around 98,000 square kilometers, which is equivalent to 3.6% of the total area of the Republic of Kazakhstan.

A pronounced continental nature characterizes the climate of northern Kazakhstan. Winters in this region are frigid, and summers are sweltering. The average air temperature in January from 2020 to 2022 was −14.1 °C, while in July it was +20.9 °C. The average annual precipitation from 2020 to 2022 was 306.25 mm, with 66% of the rainfall occurring during the warm period of the year (from April to October). The region's snow cover usually lasts about five months, from November to March.

The area under cultivation of major crops in this region averages around 4325 thousand hectares (4361 thousand hectares in 2022, 4332 thousand hectares in 2021, and 4283 thousand hectares in 2020). Cereals and legumes accounted for about 70% of the total sown area, with the share of cereals, including winter and spring wheat, making up 55%.

The region's share of gross wheat harvest (including winter and spring wheat) accounted for 23% of the total wheat production in the country for the last three years. Figure 1 illustrates the wheat fields in 2021 in the study area. According to data from the National Statistics Bureau, the average wheat yield in the North Kazakhstan Region for the period from 2020 to 2022 was 1250 kg per hectare [44].

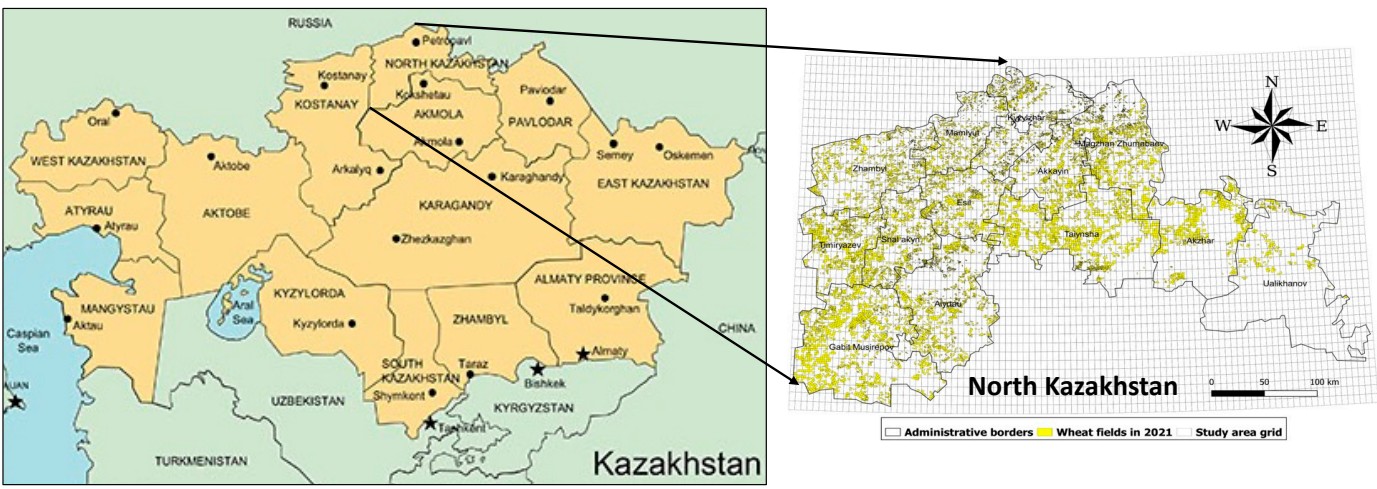

**Figure 1.** Study area and digitized wheat fields in 2021.

*2.2. Data Sources*

The CRAFT requires a large set of input data to create a project and run the simulation. The Graphical User Interface contains such sections as Gridded User Dataset, Crop mask, Crop variety, Fertilizer, Planting, Weather, Soil, etc. The reference gridded dataset in the CRAFT is the WGS84 world grid in two spatial resolutions (5 and 30 arcminutes). The schema of the area of interest was automatically generated by uploading the shapefile in three different administrative levels (larger region, region, county) of the study area, and cell IDs were assigned. The current research utilized five arcminute spatial resolution to obtain more accurate results. Further, using the digitized wheat fields for 2021, the share of wheat in each grid cell was calculated and uploaded as a crop mask layer. The date, type, quantity, and application methods of fertilizers were obtained from the recommendation developed by the North Kazakhstan Agricultural Experimental Station, "Features of cultivation of crops in the North Kazakhstan region," for 2022, taking into account an increase in the use of fertilizer in the region by 25% by 2050 [45]. Given the region's vast territory and the soil and climate characteristics, farmers cultivate wheat differently. Hence, planting methods differ from one soil and farming zone to another. Accordingly, the study area was divided into four zones (forest steppe, colo steppe, arid steppe, and small hilly), and the corresponding planting methods were assigned. Since the region receives an enormous

amount of snow during winter, rainfed agriculture predominates; thus, the current research did not consider irrigation regimes.

### 2.2.1. Weather Data

Despite the vast number of available models of the Earth system, daily climate data from two models, The Earth System Model (MPI-ESM1.2) for the High-Resolution Model Inter-comparison Project (HighResMIP) [46] and Geophysical Fluid Dynamics Laboratory's (GFDL) Earth System Model Version 4.1 (ESM4.1) [47], were used in this work. The latter has been shown to be ineffective in predicting wheat yields under the three RCP scenarios in Central Asia, most likely due to the calibration of the model using large volumes of ground-based data from North and South America. The Max Planck Institute, Germany, developed the MPI-ESM1.2. This model is widely used in the intercomparison of high-resolution models [48,49]. The model is a comprehensive tool for studying climatic and environmental processes on Earth. A feature of this model is its high spatial resolution, which allows for a more detailed study of regional climate changes. Since our goal was to assess the impact of climate change on wheat cultivation in the northern regions of Kazakhstan by 2050, the study used daily climate data from 2036 to 2065 (30-year period). Daily weather data for the North Kazakhstan region from 2015 to 2100 used for the baseline scenario are stored in the GitHub repository and are available on request.

### 2.2.2. Soil Data

Soil profile properties of the study area were obtained from the global SOILGRIDS database [50]. For each grid, soil characteristics were compiled separately and entered into the CRAFT v3.4 program. The soil analysis data were divided into six layers and varied from 5 to 200 cm depth. Moreover, soil characteristics are indicated for lower limit, upper limit drained, upper limit saturated, root growth factor, sat. hydraulic conductivity macropore, bulk density moist, organic carbon, clay, silt, coarse fraction, total nitrogen, pH in water, pH in buffer, and cation exchange capacity.

### 2.2.3. Wheat Variety

For the simulation, the local soft spring wheat variety "Shortandy 95" was chosen since this variety is widespread in the Republic of Kazakhstan and is recommended for cultivation in the Akmola and North Kazakhstan regions. According to the State Commission for Variety Testing of Agricultural Crops of the Ministry of Agriculture, the variety is medium-late, and the growing season is 95–100 days. According to approbation characteristics, it is a variety of lutescens (the ear is white, awnless, and hairless, and the grain is red). The yield in competitive variety testing for fallow was 2790 kg/ha, while the maximum yield was 4200 kg/ha [51]. The advantage of this variety is its resistance to diseases and climatic conditions, such as drought and smut. In terms of milling qualities, the weight of 1000 grains is 38–42 g, bulk density is 808 g/L, protein content is 15.7%, and raw gluten is 32.2%. A genetic model was compiled for this wheat variety using the DSSAT program. The results of this model were integrated into the CRAFT system for further simulation of the impact of climate change on wheat cultivation in 2050.

### 2.2.4. Planting Methods

Even though each farmer has their own approaches and traditions of wheat cultivation, which are passed down from generation to generation, in this study, we relied on the most optimal sowing methods recommended by the North Kazakhstan agricultural experimental station for four different soil and farming zones of the North Kazakhstan region (Table 1). Plant population at seeding varied between 250 and 400 seeds per square meter depending on the soil and climate properties of zones. The planting method (dry seeds) and plant distribution (row) were the same throughout the region, while planting depth increased from 5 cm in the forest steppe zone to 7 cm in hilly areas. As a rule, sowing should be carried out to the minimum permissible depth in a moist soil layer of at least 5 cm. When

the top layer of soil dries out, the seeding depth increases to 7–8 cm. In this regard, when deepening the seeds, it is necessary to adjust the seeding rates by 10–15% upward, which was not considered in our model. Another essential element of spring sowing agricultural technology is compliance with optimal sowing dates. If there is an expected lack of moisture, it is possible to sow 2–3 days later than usual to allow plants to use precipitation during critical phases of development effectively. However, the model needed to be more flexible to implement it. Thus, a uniform planting date was set for May 13.

**Table 1.** Planting methods in four soil and farming zones of North Kazakhstan region.

| Soil and Farming Zone | Plant Population at Seeding (PPOP), Seeds per sq. Meter | Planting Method (PLME) | Planting Distribution (PLDS) | Planting Row Spacing (PLRS), cm | Planting Depth (PLDP), cm |
|---|---|---|---|---|---|
| Forest steppe | 400 | Dry seeds | Row | 17 | 5 |
| Colo steppe | 350 | Dry seeds | Row | 17 | 5 |
| Arid steppe | 300 | Dry seeds | Row | 17 | 6 |
| Small hilly | 250 | Dry seeds | Row | 17 | 7 |

### 2.3. Methodology

After preparing the above input data in the appropriate format (administrative boundaries in SHP, weather in WTG, soil in SOL, management in tabular form, etc.), we ran the crop model and calibrated wheat yield with the observed data to obtain a spatial future wheat yield for the study area with a resolution of 5 arcminutes (Figure 2).

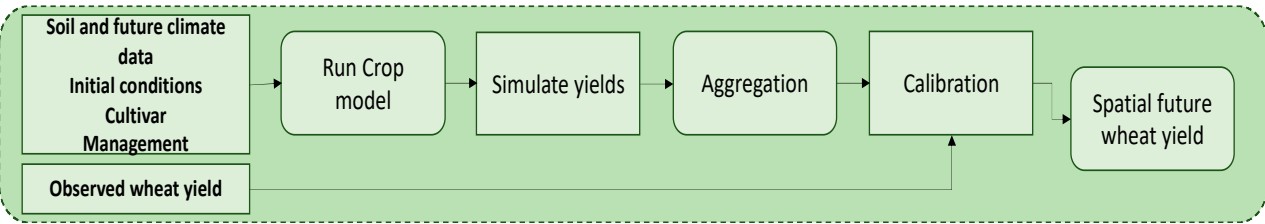

**Figure 2.** The flowchart of the future spatial wheat yield prediction.

## 3. Results

### 3.1. Wheat Yield Predictions

Wheat yield in the North Kazakhstan region was predicted under three different RCP scenarios for 2050. The average yield was calculated for the entire study area and each soil-farming zone in the region to assess the vulnerability of each zone to climate change (Table 2). According to the results, the arid steppe zone was found to be the most sensitive to an increase in carbon dioxide in the atmosphere, since the yield difference between RCPs 2.6 and 8.5 accounted for almost 110 kg per hectare (16.4%) and for 77.1 kg per hectare (10.4%) between RCPs 4.5 and 8.5. The small hilly zone was the second most vulnerable area, and the results showed an average loss of 90.1 and 58.5 kg/ha for RCPs 2.6 and 8.5 and RCPs 4.5 and 8.5, respectively. Oddly enough, the forest steppe zone was more sensitive to climate change than the colo steppe region under all three scenarios, with an average wheat loss difference of 10 kg/ha. Since we consider RCP 4.5 as a stabilization scenario and 8.5 as a negative scenario with high greenhouse gas emissions, it is crucial to estimate the potential damages in case humanity fails to achieve the Sustainable Development Goals (SDGs) until 2030. Thus, potentially, we will lose more than 10% of wheat in the arid steppe zone, 7.6% in the small hilly zone, 7.5% in the forest steppe zone, and 6% in the colo steppe zone due to climate change if the modeled RCP 8.5 scenario comes true. Overall, the average wheat yield failure in the North Kazakhstan region accounted for 25.2, 59.5, and 84.7 kg/ha for RCPs 2.6–4.5, 4.5–8.5, and 2.6–8.5, respectively.

**Table 2.** Average wheat yield in the North Kazakhstan region and each soil and farming zone under RCPs 2.6, 4.5, and 8.5 in 2050.

| Soil and Farming Zone | Average Yield RCP 2.6, kg/ha | Average Yield RCP 4.5, kg/ha | Average Yield RCP 8.5, kg/ha | Difference RCPs 2.6–4.5, kg/ha | Difference RCPs 4.5–8.5, kg/ha | Difference RCPs 2.6–8.5, kg/ha | Difference RCPs 2.6–4.5, % | Difference RCPs 4.5–8.5, % | Difference RCPs 2.6–8.5, % |
|---|---|---|---|---|---|---|---|---|---|
| Forest steppe | 847.6 | 829.7 | 766.9 | −17.8 | −62.8 | −80.7 | 2.1 | 7.5 | 10.5 |
| Colo steppe | 856.3 | 830.2 | 780.0 | −26.1 | −50.1 | −76.3 | 3.0 | 6.0 | 9.7 |
| Arid steppe | 767.7 | 736.1 | 659.0 | −31.5 | −77.1 | −108.7 | 4.1 | 10.4 | 16.4 |
| Small hilly | 798.0 | 766.4 | 707.8 | −31.6 | −58.5 | −90.1 | 3.9 | 7.6 | 12.7 |
| Total | 831.1 | 805.8 | 746.3 | −25.2 | −59.5 | −84.7 | 3.0 | 7.3 | 11.3 |

According to Table 2, the average wheat yield under RCP 2.6 was 831.1 kg/ha in the North Kazakhstan region. To illustrate the spatial variability of wheat yield, a map was compiled based on the crop model and environmental data (Figure 3). The results demonstrated the homogeneous distribution of wheat yield over the study area, since around 70% of wheat fields had 800–900 kg/ha yield under RCP 2.6. Potentially, in almost 20% of the study area, which is primarily located in the southwestern part, farmers could harvest the highest yield of 900–1000 kg/ha of wheat without any technological modernizations and genetic modifications in the mid-century. Nevertheless, eastern and southwestern parts of the study area, which are located in the arid steppe zone, had the lowest yield of 600–700 kg/ha (<3%) and 700–800 kg/ha (10%), respectively.

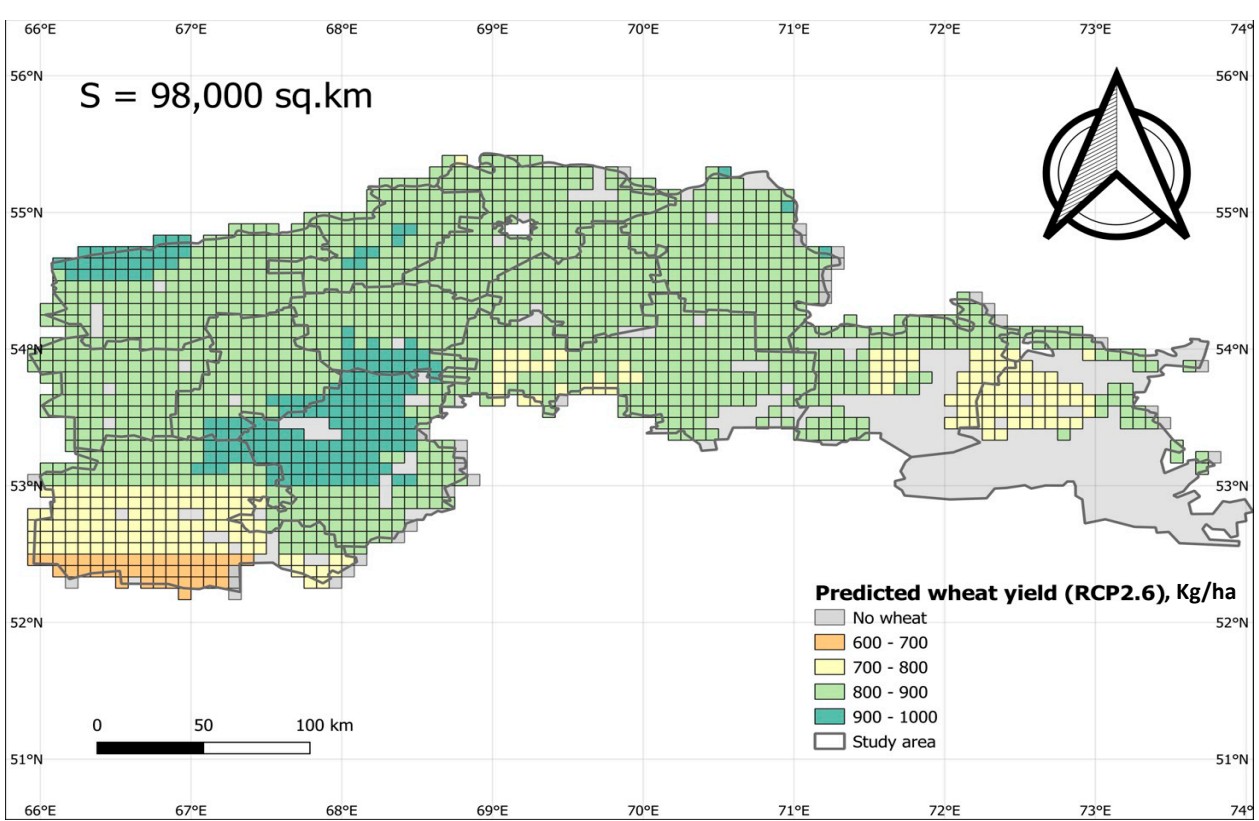

**Figure 3.** Predicted wheat yield in kg/ha under RCP 2.6 in the North Kazakhstan region for the midcentury.

The spatial distribution of wheat yield in the study area under RCP 4.5 was still relatively homogeneous, with a predominant share of wheat fields with 800–900 kg/ha (64%) yield (Figure 4). However, this does not indicate a minor difference between RCPs 2.6 and 4.5, since we observed a dramatic decrease (7%) in wheat fields with

900–1000 kg/ha yield in the southwest and a substantial increase (25%) in the territories with a lower 700–800 kg/ha yield in the east, west, and central parts of the study area. This contributed to the stable share of the area with 800–900 kg/ha yield but, overall, demonstrated an average yield decrease of 25 kg/ha in the North Kazakhstan region. The far southwestern region with the lowest yield (600–700 kg/ha) faced minor change, and its area increased by only 1%.

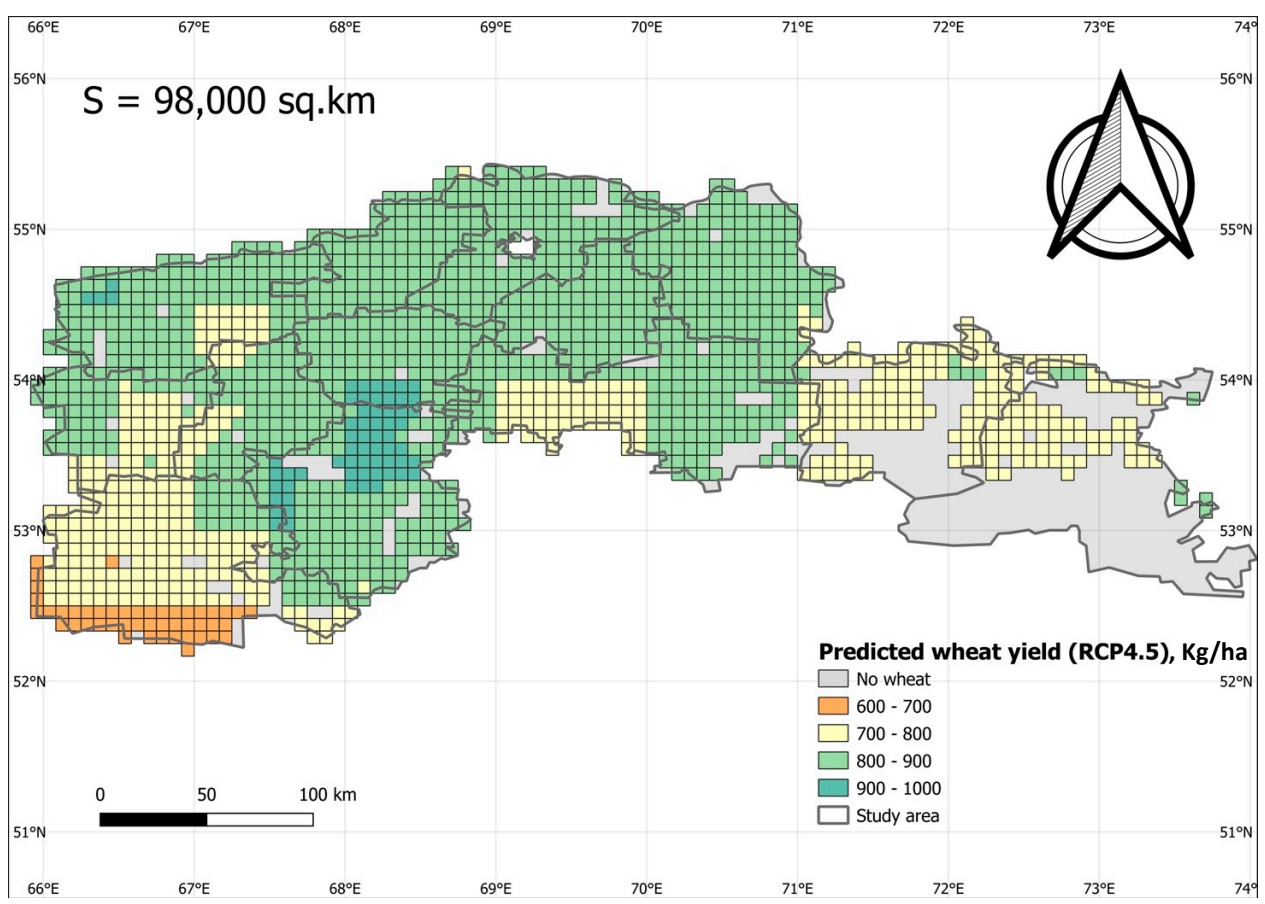

**Figure 4.** Predicted wheat yield in kg/ha under RCP 4.5 in the North Kazakhstan region for the midcentury.

The average wheat yield in the study area under RCP 8.5 decreased to 746.3 kg/ha (on average, 60 kg/ha loss compared to RCP 4.5 and 85 kg/ha compared to RCP 2.6). According to Figure 5, the negative scenario with an increased concentration of carbon dioxide in the atmosphere and, consequently, increased temperature caused the most significant harm to the wheat fields of the study area. If we observed an increase of 15% in the fields with a yield of 700–800 kg/ha under RCP scenario 4.5 relative to 2.6, then in the abovementioned figure, this area reached almost half of Northern Kazakhstan and expanded in all directions. The model also showed that agricultural fields with a yield of more than 900 kg/ha will completely disappear. Instead, farmers will harvest a maximum yield of 800–900 kg/ha in limited areas (~30%) in the central and west-central colo steppe zones. According to the results of previous scenarios, the lowest yield of 600–700 kg/ha was observed only in the southwestern part of the North Kazakhstan region, while under the RCP 8.5 scenario, it increased to 20%, covering the eastern and southwestern parts of the study area. Moreover, a new class appeared on the map, showing meager yields of 500–600 kg/ha in the far southwestern part, and its area was equal to 4%.

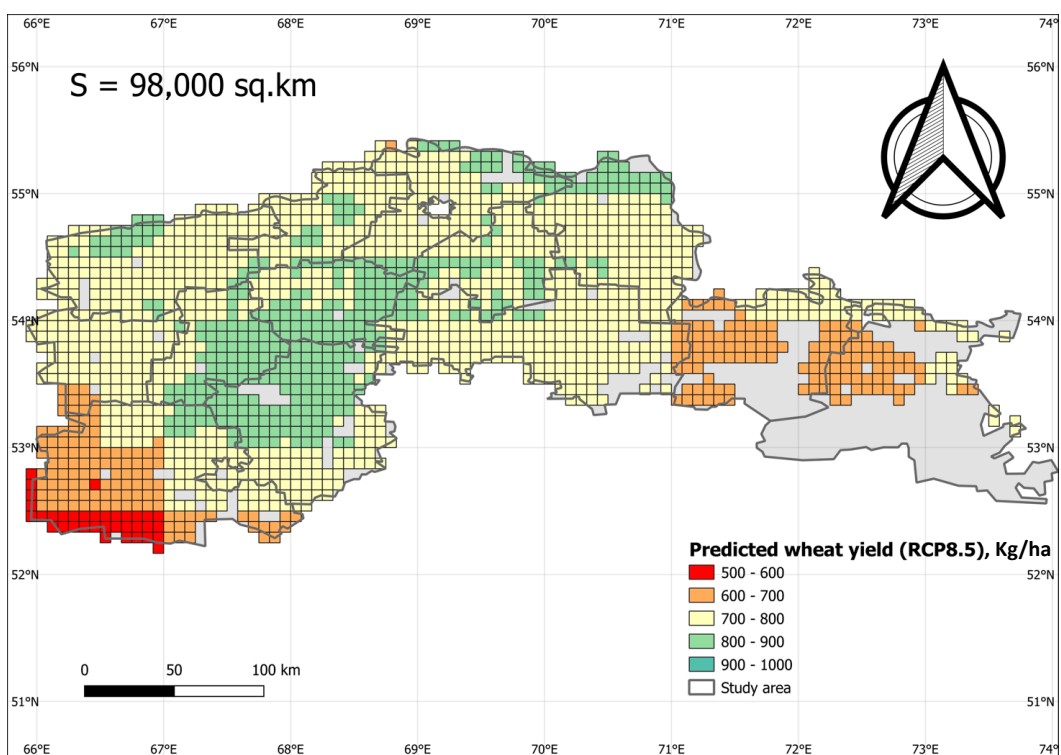

**Figure 5.** Predicted wheat yield in kg/ha under RCP 8.5 in the North Kazakhstan region for the midcentury.

Predicted Wheat Yield Difference between RCPs 2.6, 4.5, and 8.5

According to Table 2, the average wheat yield failure in the North Kazakhstan region accounted for 25.2, 59.5, and 84.7 kg/ha for RCPs 2.6–4.5, 4.5–8.5, and 2.6–8.5, respectively. To demonstrate the spatial variations, maps were compiled showing differences in wheat yield between three scenarios of RCP with a spatial resolution of 5 arcminutes (Figure 6). Nevertheless, since the climate data received from the Earth System Model version 1.2 from the Max Planck Institute had a 30 arcminute resolution interpolated to 5 arcminutes, the produced maps contained relatively similar yield difference results inside those square shapes. The highest predicted wheat yield difference between RCPs 2.6 and 4.5 was observed in the arid steppe zone located in the eastern part of the study area and east of the colo steppe zone, which varied from 40 to 60 kg/ha loss of wheat, while the northern and southwestern parts (predominantly in the forest steppe zone) faced only minor changes of 0–20 kg/ha loss. Dramatical yield decrease in the arid steppe zone could be explained by relatively low soil fertility (saline tertiary sediments), increased level of evaporation due to high air temperatures and strong steppe winds which exceed the amount of precipitation, and the low xerophilicity of the wheat varieties used. However, a comparison of RCP 4.5–8.5 results illustrated dramatic wheat yield failure (60–80 kg/ha) across the forest steppe and small hilly zones and severe yield anomalies (80–110 kg/ha) in the arid steppe zone both in the southwest and east parts of the study area. The most red-colored map (53% of the area had 80–130 kg/ha loss) was the RCP 2.6–8.5 comparison map. Severe yield failure was observed along all edges and the western and eastern parts of the study area that primarily occupied the arid steppe, small hilly, and forest zones. As a result, it was found that the colo steppe zone in the central part of the study area has a higher resistance to climate change (20–60 kg/ha damage) when growing spring wheat using current technologies and varieties. This is because the colo steppe zone is predominantly located on the floodplain of the Ishim paleochannel, which has highly fertile soil and moisture reserves.

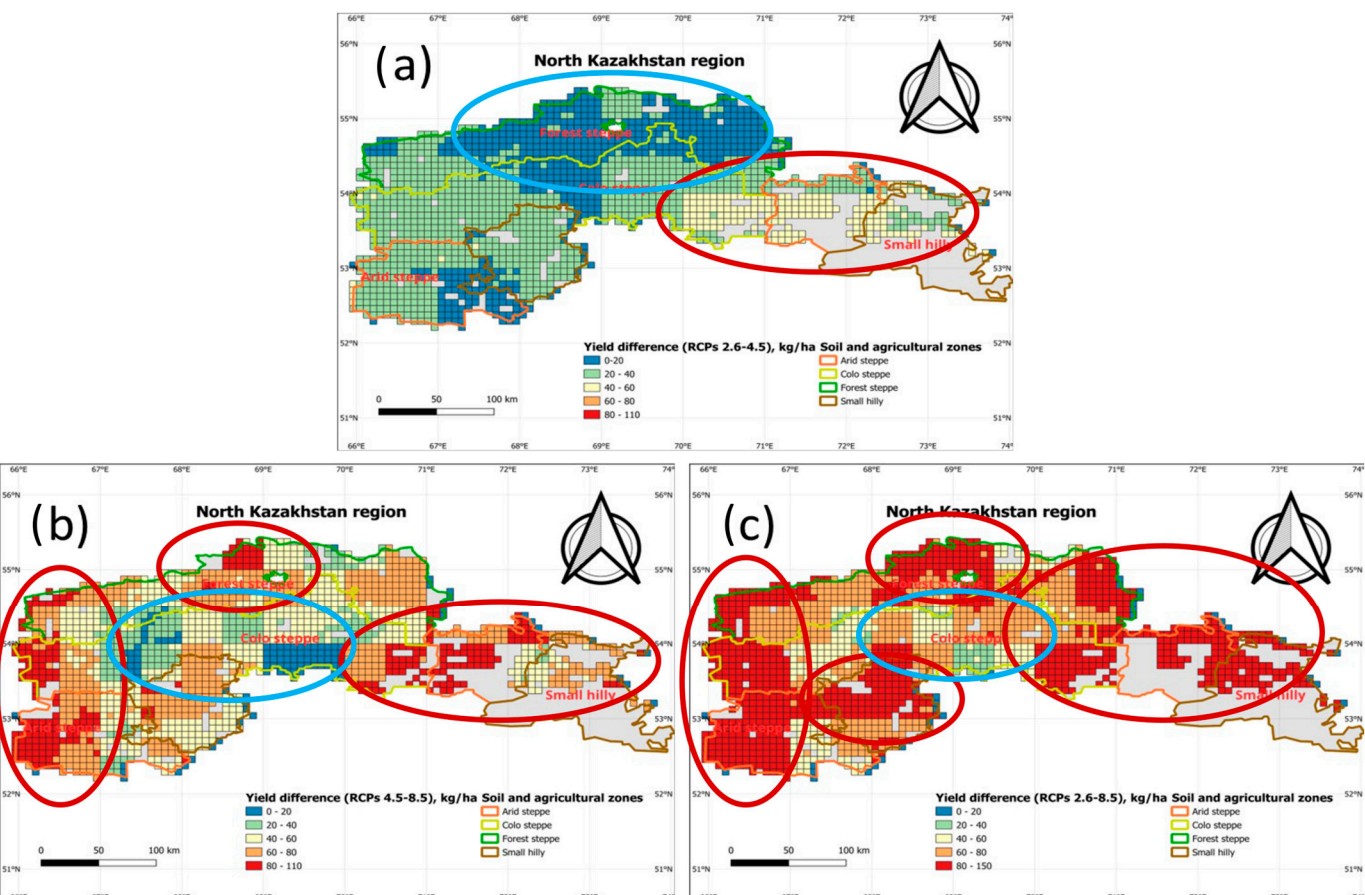

**Figure 6.** Predicted wheat yield difference in kg/ha between RCPs 2.6–4.5 (**a**), 4.5–8.5 (**b**), and 2.6–8.5 (**c**) in the North Kazakhstan region. Blue and red circles indicate favorable and unfavorable areas, respectively.

## 4. Discussion

There has been a massive demand for modern geographic (spatial) yield forecasting tools in recent decades and a corresponding rise in the accessibility of global geospatial data sets from several data-gathering sources. Different tools utilizing various techniques have been effectively assisted by statistical seasonal weather predictions [52–55]. These involve utilizing crop models in conjunction with seasonal precipitation estimates based on GCMs (General Circulation Models) [56,57] and historical data [58,59]. Although the CRAFT was designed with an integrated Climate Predictability Tool for seasonal forecasts, it is not limited to this. It allows one to enter external climate data from different sources. Unlike its predecessors (the Agricultural mad Environmental Geographic Information System for Windows (AEGIS/WIN) [60], the Joint Research Centre's Biophysical Models Applications (BioMA) framework [61], and the GIS-based Environmental Policy Integrated Climate (GEPIC) model [55]), the CRAFT allows multiple pre-installed crop models such as DSSAT, APSIM, and SARRA-H to be run, thus being a successful product for predicting both seasonal and long-term crop yields with a sufficiently high spatial resolution.

The above results Indicate that, by combining the DSSAT crop model and long-term climate predictions, the CRAFT can successfully assess the impact of climate change on agricultural production under different climate scenarios. Considering 2 million hectares of wheat fields in the North Kazakhstan region and the average wheat yield of 1250 kg/ha between 2020 and 2022 [44], the world market receives 2.5 billion kilograms or 2.5 million tons of wheat annually. However, even if humanity can reduce carbon dioxide emissions into the atmosphere by zero and methane by half by 2100 (RCP 2.6), global temperatures will still continue to rise naturally to about 2 degrees, and according to our study based

on spatial simulations using CRAFT, it will cause a loss of 837,800 tons of wheat or USD 167 million (the market price of wheat varies significantly due to different events worldwide, but for the calculation, an average price of USD 200 per ton was determined) annually in the mid-century. There is a debate among scientists worldwide if rising temperatures and a higher concentration of carbon dioxide in the atmosphere will boost or decrease wheat yield in different continents and soil and climate conditions. Even though the experiments conducted in the laboratory-chamber, greenhouse, and open- and closed-top field chamber demonstrate a positive effect of elevated $CO_2$ on wheat yield [62–66], overall, direct and indirect effects of climate change will result in yield and nutritional quality decrease [67–71]. Many scholars agreed that the RCP 4.5 is the most probable baseline scenario [72,73], under which the simulation in the North Kazakhstan region demonstrated wheat failure of 25.2 kg/ha compared to RCP 2.6 or 888,400 tons (USD 177 million) relative to today's yield. Under the improbable but still possible RCP 8.5 scenario, the world market will receive around 1 million tons of wheat (USD 201 million) less annually, which may lead to regional food disasters. Nevertheless, the model and calculations we used do not consider future technological modernization and genetic modification aimed at adapting to climate change. Moreover, the limited availability of high-resolution climate data might cause spatial wheat yield bias.

**5. Conclusions**

In this study, the effects of three distinct Representative Concentration Pathways with 2.6 W/m$^2$, 4.5 W/m$^2$, and 8.5 W/m$^2$ on wheat production in the North Kazakhstan region were assessed using the DSSAT crop model spatial application using the CRAFT and daily forecasted climate data from the MPI-ESM1.2. As a result, the arid steppe zone was found to be the most sensitive to an increase in greenhouse gases in the atmosphere, since the yield difference between RCPs 2.6 and 8.5 accounted for almost 110 kg/ha (16.4%) and 77.1 kg/ha (10.4%) between RCPs 4.5 and 8.5, followed by the small hilly zone with an average loss of 90.1 and 58.5 kg/ha for RCPs 2.6–8.5 and RCPs 4.5–8.5, respectively. It was found that by the mid-century, potentially, we will lose more than 10% of wheat in the arid steppe zone, 7.6% in the small hilly zone, 7.5% in the forest steppe zone, and 6% in the colo steppe zone due to climate change if the modeled RCP 8.5 scenario comes true. The spatial variation maps for RCPs 2.6 and 4.5 demonstrated a homogeneous distribution of wheat yield over the study area since around 70% of wheat fields produced from 800 to 900 kg/ha yield. A comparison of RCP 4.5–8.5 results illustrated dramatic wheat yield failure (60–80 kg/ha) across the forest steppe and small hilly zones and severe yield anomaly (80–110 kg/ha) in the arid steppe zone both in the southwest and east parts of the study area. Lastly, it was found that the colo steppe zone in the central part of the study area has a higher resistance to climate change (20–60 kg/ha damage) when cultivating spring wheat using current technologies and wheat varieties.

Overall, the CRAFT v3.4 and DSSAT 4.7.5 software application program combined with the Earth System Model's climate predictions showed great potential in assessing climate change effects on wheat yield under different climate scenarios in the North Kazakhstan region. We believe that the results obtained will be helpful during the development and zoning of modified, drought-resistant wheat varieties and the cultivation of new crops in the region. Further work is required to improve simulation models and provide available and accurate environmental data at a high spatial resolution to consider the full range of factors affecting wheat production (e.g., flexible sowing date for each year, which changes automatically to effectively use rainfall during critical phases of crop development instead of fixed ones).

**Author Contributions:** Conceptualization, Z.T., A.R. and V.S.; methodology, Z.T., V.S. and G.H.; software, A.R. and A.Z.; formal analysis, I.T.; investigation, Z.T. and S.T.; resources, I.T. and F.Y.; data curation, A.R. and Z.T.; writing—original draft preparation, S.T., A.R., A.Z. and Z.T.; writing—review and editing, Z.T., V.G. and A.I.; visualization, Z.T.; supervision, I.T., V.S. and G.H.; project administration, I.T.; funding acquisition, I.T. and F.Y. All authors have read and agreed to the published version of the manuscript.

**Funding:** The research was financially supported by grant IRN BR10865099 from the Ministry of Agriculture of the Republic of Kazakhstan from 2021 to 2023.

**Institutional Review Board Statement:** Not applicable.

**Data Availability Statement:** Daily weather data for the North Kazakhstan region from 2015 to 2100 are stored in the GitHub repository and are available on request.

**Conflicts of Interest:** The authors declare no conflicts of interest.

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
