# Peer review of "Comparison of Climate Change Effects on Wheat Production under Different Representative Concentration Pathway Scenarios in North Kazakhstan"

_sustainability, doi:10.3390/su16010293_

Round 1

Reviewer 1 Report

Comments and Suggestions for Authors

I suggest to change the title to “ Comparison of climate change effects on wheat production under different RCP scenarios in Norht Kazakhstan”

L15: Rewrite the first statement in the abstract.

L19: replace “understudied” with appropriate word, for example, “…research work in Central Asia is limited”.

L39: change “symphony” to an appropriate word.

L43: change “intersection” to interaction.

L44: “abstract concern” is not appropriate..

The introduction is long and wordy and has to be shortened and condense.

The discussion section did not provide interpretation of results, and only few previous findings were used.

The conclusion section is long, and needs to be strengthened.

The entire manuscript English needs to be edited by native English speaker.

Comments on the Quality of English Language

The entire manuscript English needs to be edited by native English speaker.

Author Response

Thank you for your comments and suggestions. Please find the response attached below.

Reviewer 2 Report

Comments and Suggestions for Authors

Review Comments for Manuscript ID sustainability- 2742674

The manuscript offers an insightful and comprehensive overview of the impact of climate change on wheat cultivation in the North Kazakhstan region. The authors have examined various aspects, including the selection of wheat varieties, planting methods, and the utilization of the CRAFT toolbox for modeling and predicting wheat yield under different Representative Concentration Pathways (RCPs).

After careful consideration, I would like to recommend this manuscript for publication in Sustainability (ISSN 2071-1050), provided the authors address major revisions. These suggestions are outlined below:

Abstract:

1.      It is essential to clarify the meaning of acronyms, especially those that may not be universally known. In this context, RCP refers to "Representative Concentration Pathways." I recommend incorporating this expansion in the first mention of RCP in the abstract to ensure that readers unfamiliar with the acronym gain an immediate understanding of its significance.

2.      It would be beneficial to explicitly state the geographic boundaries or specific regions within North Kazakhstan that were included in your analysis. Providing this information early in the abstract would offer readers a clear context and aid in the interpretation of your subsequent findings. Thus, it is recommended to specify that, for example, the study area was divided into four zones (forest-steppe, colo steppe, arid steppe, and small hilly)…

3.      The reported figures for wheat yield failure could be presented in a manner that facilitates easy comparison. Consider expressing these values as percentages, as this format may offer a clearer perspective. This adjustment would provide a more accessible context for readers to comprehend the significance of the projected reductions.

Introduction:

4.    The language employed in the introduction may be considered unconventional in an academic context. I recommend that the authors review and modify the wording to align it more closely with academic conventions.

5.    It would be beneficial to create a logical framework or diagram illustrating the study procedures undertaken by the authors. This diagram should encompass and interconnect various elements, such as meteorological data, the wheat variety employed in the simulation, planting methods, among others. This visual aid would enhance the comprehension of the study and could be complemented by the flowchart presented in Figure 2.

6.    The authors should elucidate the underlying reasons attributed to the significant difference observed in the arid steppe zone, both from RCPs 2.6 to 4.5 and from 4.5 to 8.5. It would be insightful for the authors to incorporate a causal explanation.

7.    Similarly, following the same line of inquiry, it would be valuable for the authors to provide a contextual understanding of causality to elucidate the heightened resilience to climate change observed in the central part of the colo steppe zone.

Sincerely,

Comments on the Quality of English Language

I have recommended to the authors to enhance certain aspects related to English language usage, as highlighted in the 'Comments and Suggestions for Authors.

Author Response

(The authors gave the same response as above.)

Reviewer 3 Report

Comments and Suggestions for Authors

Authors have studied the effect of 3 different RCP on the wheat production in North Kazakhstan using CRAFT model. Authors have written the manuscript really well and all the results and predictions are explained well. Authors are  advised to consider the following minor suggestions/corrections to revise their manuscirpt.

1. In the introduction section, it would be good to include any data available on Kazakhstan, explaining how the wheat yield or any crop yield is affected overtime due to different parameters.

2. Please provide links to the global databases used in the study. For example: SOILGRIDS

3. Line 273: what is nature 808g/l?

4. Line 405: please elaborate GCM?

5. Line 421 and 422: check the sentence once again and correct it

After the incorporation of the minor corrections, the manuscript can be accepted for publication.

Author Response

(The authors gave the same response as above.)

Reviewer 4 Report

Comments and Suggestions for Authors

Dear authors,

-Please explain the RCPs 2.6, RCPs 4.5 and 8.5. For your information to be relevant there needs to be a clear understanding of the scenarios (in detail).

-The first sentences in the introduction are poetic. Though they are correct I suggest rephrase them in order to sound scientific.

-line 15, line 64- please rephrase

-line 197- move on the next page

- line 246- please make this discussion more obvious (why the system isn't suitable for predicting yields)

-Line 319- this information will be much easier to see in a graph.

-Line 345- please rephrase this part in order to be more clear

Author Response

(The authors gave the same response as above.)

Round 2

Reviewer 2 Report

Comments and Suggestions for Authors

Dear Authors,

I want to express my appreciation for your consideration of the feedback provided during the review of your manuscript.

Best regards,